# Isolation of Primary Human Saphenous Vein Endothelial Cells, Human Internal Thoracic Artery Endothelial Cells, and Human Adipose Tissue-Derived Microvascular Endothelial Cells from Patients Undergoing Coronary Artery Bypass Graft Surgery

**DOI:** 10.3390/ijms26189217

**Published:** 2025-09-21

**Authors:** Daria Shishkova, Yulia Yurieva, Alexey Frolov, Vera Matveeva, Evgenia Torgunakova, Victoria Markova, Anastasia Lazebnaya, Anton Kutikhin

**Affiliations:** Department of Experimental Medicine, Research Institute for Complex Issues of Cardiovascular Diseases, 6 Barbarash Boulevard, 650002 Kemerovo, Russia; shidk@kemcardio.ru (D.S.); markyo@kemcardio.ru (Y.Y.); frolav@kemcardio.ru (A.F.); matvvg@kemcardio.ru (V.M.); torgea@kemcardio.ru (E.T.); markve@kemcardio.ru (V.M.); lazeai@kemcardio.ru (A.L.)

**Keywords:** coronary artery disease, coronary artery bypass graft surgery, endothelial cell isolation, saphenous vein, internal thoracic artery, subcutaneous adipose tissue, microvessels, immunophenotyping, endothelial cell proliferation, angiogenesis

## Abstract

Primary human endothelial cells represent an essential tool to model endothelial dysfunction and to screen interventions for its treatment. Here, we developed a protocol for the synchronous isolation of primary human saphenous vein endothelial cells (HSaVEC), human internal thoracic artery endothelial cells (HITAEC), and human microvascular endothelial cells (HMVEC) from SV and ITA utilized as conduits during coronary artery bypass graft surgery and from subcutaneous adipose tissue excised while providing an access to the heart. Treatment by collagenase type IV and magnetic separation with anti-CD31-antibody-coated beads ensured relatively high efficiency of the isolation (≈60% for HSaVEC, ≈50% for HITAEC, and ≈20% for HMVEC) and high purity (≥99%) of isolated ECs within ≈2 weeks (HSaVEC), ≈2–3 weeks (HITAEC), and ≈3–4 weeks (HMVEC). A colorimetric assay of cell viability and proliferation, as well as real-time bioimpedance monitoring using the xCELLigence instrument, demonstrated high proliferative activity in HSaVEC, HITAEC, and HMVEC, whilst the in vitro tube formation assay indicated their angiogenic potential. The isolation of HSaVEC, HITAEC, and HMVEC from patients undergoing coronary artery bypass graft surgery is a promising option to investigate endothelial heterogeneity, to interrogate endothelial responses to various stresses, and to pinpoint the optimal approaches for restoring endothelial homeostasis, thereby reproducing them within the bedside-to-bench-to-bedside concept.

## 1. Introduction

The endothelial cell (EC) lining of blood vessels is indispensable for preventing thrombosis, regulating vascular tone for controlling blood pressure, and sustaining basal levels of pro-inflammatory cytokines in human blood [1,2,3,4]. ECs secrete key angiocrine factors such as vasoactive molecules (e.g., nitric oxide, prostacyclin, endothelin-1, thromboxane A2), angiogenic and pro-inflammatory molecules (e.g., angiopoietin-2, monocyte chemoattractant protein 1, macrophage and granulocyte colony-stimulating factors, macrophage migration inhibitory factor, interleukin-6, interleukin-8, interleukin-32, and chemokine (C-X-C motif) ligand 1), and pro-fibrotic mediators (such as transforming growth factor beta) which act as molecular cues for vascular smooth muscle cells, pericytes, and immune cells. Further, ECs orchestrate angiogenesis and regeneration, thus playing a pivotal role in post-ischemic tissue repair [1,2,3,4]. Likewise, vascular insufficiency observed in diabetes mellitus [5,6,7,8], chronic kidney disease [7,8], COVID-19 [9], or after full-thickness burns [10] significantly inhibits recovery after injuries. Systemic disorders such as ischemia [11,12], dyslipidemia [13,14], obesity [15,16,17], hyperglycemia [14,18,19], azotemia [20,21,22], uremia [20,21,22], and hyperphosphatemia [23,24], which are frequently observed in patients with atherosclerotic vascular disease, diabetes mellitus, and chronic kidney disease, stimulate pathological activation of ECs accompanied by an increased release of pro-inflammatory cytokines into the circulation and by a migration of leukocytes into the blood vessel wall [1,2,3,4]. Together, these factors promote the development of chronic low-grade inflammation that enhances: (1) arterial hypertension through the reduced production of endothelial vasodilators [4]; (2) arterial stiffness by inducing phenotypic switch of vascular smooth muscle cells which leads to decreased contractility and elevated synthesis of collagen and matrix-degrading enzymes (matrix metalloproteinases, cathepsins, and metalloendopeptidases) [25,26,27,28,29]; (3) inflammaging-driven frailty syndrome defined as accelerated age-related physical and cognitive deficit [30,31,32,33,34]. Specific triggers such as severe acute respiratory syndrome coronavirus 2 (SARS-CoV-2) infection might induce a pro-thrombotic state of the ECs characterized by an elevated production of von Willebrand factor multimers and plasminogen activation inhibitor, along with a diminished synthesis of urokinase- and tissue-type plasminogen activator, together provoking arterial, venous, and microvascular thrombosis and its life-threatening complications [35,36,37,38,39,40,41,42,43,44]. A large number of ECs exist in the human body (ranging from 1 to 60 trillion, according to different estimates), and their direct contact with the bloodstream underscores the pathophysiological and clinical importance of endothelial dysfunction, emphasizing the need for the timely implementation of efficient lifestyle and pharmacological approaches for its treatment [1,2,3,4,45,46].

Arterial, venous, and microvascular ECs show significant molecular and functional heterogeneity because of distinct biomechanical conditions (i.e., shear stress, cyclic strain, and blood pressure), different partial pressure of oxygen and carbon dioxide, and variable extracellular matrix stiffness in the respective blood vessels [47,48,49,50,51]. Endothelial heterogeneity implies the use of distinct EC lineages for studying vascular diseases [47,48,49,50,51]. As such, arterial ECs are widely employed in atherosclerosis research [52,53,54,55,56,57,58,59,60], venous ECs are utilized to investigate the development of thrombosis [61,62,63], and microvascular ECs are required to interrogate microcirculatory dysfunction [64,65,66]. In line with population aging, the prevalence of comorbid conditions (i.e., diabetes mellitus, chronic kidney disease, and chronic obstructive pulmonary disease) in patients with cardiovascular disease has increased in recent years [67,68,69,70], demanding fine-tuning of the routine disease models to the current pathophysiological scenarios. Isolation of arterial, venous, and microvascular ECs during the patient’s lifetime offers an opportunity to model endothelial dysfunction in vitro and to screen multiple treatment regimens for subsequent delivery of an optimal therapeutic approach to the same patient in order to reach the favorable outcome.

Coronary artery bypass graft (CABG) surgery, which is conducted in patients with severe coronary artery disease in order to restore the blood flow, implies grafting of both the internal thoracic artery and the saphenous vein for employing them as conduits to bypass the stenosis [71,72]. Excessive fragments of these blood vessels and small amounts of vascular-rich subcutaneous adipose tissue, which are excised to provide access to the heart, can then be applied for the isolation of primary arterial, venous, and microvascular ECs. In this clinical setting, human saphenous vein endothelial cells (HSaVEC), human internal thoracic artery endothelial cells (HITAEC), and human adipose tissue-derived microvascular endothelial cells (HMVEC) are relatively intact as none of these blood vessels (harnessed as conduits for CABG surgery) is affected by atherosclerosis, stenosis, or thrombosis in these cases, although the patient still suffers from the respective comorbid conditions. Hence, these ECs might have good proliferative and angiogenic properties while still harboring the pathogenetic features from multimorbid patients. Importantly, HSaVEC, HITAEC, and HMVEC can be isolated from the same patient, which gives an advantage to avoid inter-individual differences related to the donors but not EC types. Although in vitro modeling is unable to fully recapitulate biochemical and biomechanical in vivo conditions, EC culture in serum-free and nutrient-deprived medium permits the analysis of an endothelial response and the extraction of the complete set of endothelial-derived proteins at starvation corresponding to a post-ischemic metabolic stress.

In this study, we report a detailed protocol for the isolation of HSaVEC, HITAEC, and HMVEC from 40 patients who underwent CABG surgery. Having employed enzymatic degradation of the basement membrane by collagenase type IV and positive magnetic separation with anti-CD31-antibody-coated beads, we were able to reach relatively high efficiency of the isolation (≈60% for HSaVEC, ≈50% for HITAEC, and ≈20% for HMVEC) and high purity (≥99%) of isolated ECs with high proliferative and angiogenic activity. The duration between the isolation and obtaining pure culture of the ECs in the standard T-75 flask (≥3 × 10^6^ cells) was ≈2 weeks for HSaVEC, ≈2–3 weeks for HITAEC and ≈3–4 weeks for HMVEC. The isolated ECs fully retained proliferation capability and endothelial phenotype over ≥5 passages and hence can be used for the extended expansion to apply a variety of molecular biology and cell biology techniques. We suggest that primary vascular ECs isolated by our approach can be applied in research exploring venous physiology (HSaVEC), investigating the molecular basis of endothelial resilience (HITAEC), and studying microcirculatory dysfunction (HMVEC) in the real-world patient setting. As all of these ECs are isolated during the patient’s lifetime, they can be used for the modeling of endothelial dysfunction and for the selection of the optimal options for its treatment to further apply these therapies to the patient in accordance with the bedside-to-bench-to-bedside concept.

## 2. Results

We have successfully isolated HSaVEC, HITAEC, and HMVEC from 24/40 (60.00%), 19/40 (47.50%), and 7/40 (17.50%) patients who underwent CABG surgery (Table 1). Most patients were of male gender (31/40, 77.50%), and significant proportions of the patients were diagnosed with arterial hypertension (39/40, 97.50%), dyslipidemia (13/40, 32.50%), obesity (i.e., body mass index ≥ 30 kg/m^2^, 12/40, 30.00%), diabetes mellitus (14/40, 35.00%), chronic kidney disease (21/40, 52.50%), chronic obstructive pulmonary disease (4/40, 10.00%), cerebrovascular disease (10/40, 25.00%), and peripheral artery disease (8/40, 20.00%, Table 1). The average age of the patients was 67.3 ± 6.5 years.

On days 3–4, we observed single polygonal or slightly elongated ECs, or EC colonies with a typical cobblestone appearance, including 12–15 (HSaVEC), 8–14 (HITAEC), or 7–10 (HMVEC) cells that confirmed successful adhesion and adaptation to the cell culture conditions (Figure 1, Figure 2 and Figure 3, Table 2). On average, HSaVEC, HITAEC, and HMVEC formed 15–17, 8–10, and 3–5 colonies per donor, respectively (Table 2). HSaVEC required 1–2 positive immunomagnetic separations with anti-CD31-antibody-coated beads and formed a cobblestone monolayer in T-25 flasks after 8–10 days of culture (Figure 1, Table 2), whilst HITAEC needed 3–4 separations and grew to a monolayer within 10–12 days (Figure 2, Table 2). Yet, HMVEC demanded 5–6 separations and 14–16 days to reach confluence because of a smaller number of colonies (Figure 3, Table 2). After the passage of ECs into the T-75 flasks, a cobblestone monolayer of HSaVEC, HITAEC, and HSaVEC in T-75 flasks was, respectively, obtained at the 14–16, 18–20, and 24–26 days of culture following the isolation procedure (Figure 3, Table 2). Immunophenotyping of HSaVEC, HITAEC, and HMVEC by means of flow cytometry found ≥99% CD31^+^CD146^+^CD90^−^ cells, where expression of CD31 and CD146 reflected endothelial specification and the absence of CD90 expression excluded the fibroblast identity (Figure 1, Figure 2 and Figure 3). CD31^+^CD146^+^CD90^+^ phenotype, which could indicate activated ECs, has not been detected in primary isolates or after the immunomagnetic reselections. Collectively, these parameters indicated a canonical endothelial phenotype, suggesting high proliferation and viability of the isolated HSaVEC, HITAEC, and HMVEC.

To verify the endothelial phenotype of HSaVEC, HITAEC, and HMVEC, we performed confocal microscopy following the immunostaining for platelet endothelial cell adhesion molecule 1 (CD31/PECAM1) and von Willebrand factor enclosed in Weibel-Palade bodies within the EC cytoplasm (Figure 4). In all cases, HSaVEC, HITAEC, and HMVEC formed tight contacts within the monolayer and displayed intense CD31/PECAM1 signal on the cell membrane as well as von Willebrand factor signal in the cytoplasm (Figure 4).

We next evaluated the proliferation rate of HSaVEC, HITAEC, and HMVEC over 72 h using cell viability and proliferation assay and measurements of bioimpedance by the xCELLigence real-time cell analysis instrument. In keeping with the phase contrast microscopy observations, HSaVEC, HITAEC, and HMVEC showed a stable increment in the ratio of optical density at 570 and 605 nm wavelength (OD_570_/OD_605_, Figure 5) and cell index (the relative change in measured impedance, Figure 6) along the 24, 48, and 72 h time points. In contrast to many cell types, ECs contribute to impedance by both occupying the electrode surface and establishing tight intercellular contacts. Yet, the combination of phase contrast microscopy, colorimetric cell viability and proliferation assay, and bioimpedance measurements collectively provided evidence for the stable proliferation of HSaVEC, HITAEC, and HMVEC.

As angiogenesis represents a vital function of ECs, we further tested the ability of HSaVEC, HITAEC, and HMVEC to form capillary-like tubes in a 3D gel matrix imitating the endothelial basement membrane (Figure 7). Seeding of HSaVEC, HITAEC, and HMVEC into the solubilized basement membrane preparation extracted from Engelbreth-Holm-Swarm mouse sarcoma for 24 h resulted in the formation of capillary-like tubes, confirming the angiogenic competence of the ECs (Figure 7).

We next assessed the ability of HSaVEC, HITAEC, and HMVEC to proliferate from the isolation of the pure culture (≥99% CD31^+^CD146^+^CD90^−^ cells) before undergoing senescence. Although the number of senescent cells increased from passage 7 to passage 11, all of these EC types formed a monolayer for at least 11 passages, indicating a good expansion potential (Figure 8).

To better characterize the molecular profiles of HSaVEC, HITAEC, and HMVEC after their pro-inflammatory activation, we incubated these ECs with tumor necrosis factor alpha (TNF-α) for 24 h and collected RNA, total protein, and cell culture supernatant. Reverse transcription-quantitative polymerase chain reaction (RT-qPCR) profiling indicated upregulation of *ICAM1*, *IL6*, and *CCL2* genes upon TNF-α stimulation (Table 3). In line with RT-qPCR results, Western blotting confirmed the overexpression of intercellular cell adhesion molecule 1 (ICAM1) in the lysate of ECs exposed to TNF-α (Figure 9), whilst enzyme-linked immunosorbent assay (ELISA) showed an elevated release of interleukin-6 and monocyte chemoattractant protein 1/C-C motif chemokine ligand 2 (MCP-1/CCL2) by TNF-α-treated ECs (Figure 10).

## 3. Discussion

Here, we developed a protocol for isolating HSaVEC, HITAEC, and HMVEC from patients undergoing CABG surgery. Significant amounts of the mentioned ECs (≥3 × 10^6^ cells, equal to the confluent T-75 flask) were obtained within ≈2 weeks in case of HSaVEC, ≈2–3 weeks in case of HITAEC, or within ≈3–4 weeks in case of HMVEC. The presented approach to obtain primary ECs holds several advantages: (1) opportunity to collect arterial, venous, and microvascular ECs from the same patient without any inherited heterogeneity between distinct EC specifications; (2) a significant yield of highly proliferative and angiogenic ECs which retain endothelial phenotype for ≥5 passages; (3) exposure of ECs to systemic triggers of endothelial dysfunction in combination with the location in the intact blood vessels which are not affected by vascular lesions; (4) isolation of the ECs during the patient’s lifetime with the possibility to fine-tune therapeutic approaches for restoring endothelial homeostasis in vitro and for their subsequent translation to the same patient suffering from endothelial dysfunction. These benefits make this approach superior to the isolation of carotid artery ECs from patients undergoing carotid endarterectomy due to atherosclerosis, aortic ECs from patients with thoracic or abdominal aneurysms, pulmonary artery ECs from patients with pulmonary artery hypertension, HSaVEC from patients undergoing surgical removal of varicose veins (i.e., phlebectomy), aortic valve ECs during heart valve replacement because of aortic stenosis, or tumor-associated ECs from patients with cancer. In these cases, ECs are isolated from diseased blood vessels or heart valves and therefore cannot be considered intact, although their proliferative capabilities can be similar to HSaVEC, HITAEC, and HMVEC isolated from patients undergoing CABG surgery.

An alternative approach to isolate ECs includes isolation of human umbilical vein endothelial cells (HUVEC), which are abundant in the umbilical cord collected upon childbirth [73,74,75]. Such protocols are described in detail elsewhere and are relatively standardized, simplifying the procedure and facilitating its broad implementation [76,77,78,79,80,81]. Advantages of using HUVEC also include their ability to provide about nine useful passages, allowing expansion of early generations. Yet, we have shown that HSaVEC, HITAEC, and HMVEC can also proliferate to form a monolayer for at least 11 passages, although undergoing a gradual senescence from passage 7 to passage 11. Currently, HUVEC are the most frequently used EC type for the study of biomaterials [75], cardiovascular disease [73,82], and endothelial physiology [73,82,83,84,85,86]. However, HUVEC are neonatal ECs that do not fully correspond to ECs isolated from adults, in particular patients with systemic disorders such as atherosclerosis. Studies of hematopoietic stem cells [87], fibroblasts [88], and ECs [89,90,91,92] demonstrated that somatic cells accumulate genomic and epigenetic alterations with age. Further, the proliferative capacity of peripheral blood-derived endothelial colony-forming cells isolated from young adults born preterm was lower in comparison with those born full-term [93]. These studies suggest that ECs retain their molecular and functional features at least for several in vitro passages [89,90,91,92,93]. Molecular heterogeneity between neonatal and adult venous ECs and its functional consequences remain unclear, and unbiased comparison of these cells seems complicated, as adult organism lacks the umbilical vein, and it would take decades to isolate venous ECs from the same donor at childbirth and in the adult state. The comparison of HUVEC with HSaVEC isolated from patients undergoing CABG surgery can provide valuable data, as in this case, both EC types are obtained from intact veins. Yet, this approach does not exclude inter-individual variability of ECs.

Another source of ECs that is commonly employed by commercial institutions is cadaveric tissues. Although the isolation of cadaveric ECs is a promising approach to obtain significant amounts of ECs from multiple blood vessels, it does not imply further clinical applications of these EC types and limits their use to basic to translational science. Cadaveric ECs are also affected by the postmortem changes, which become evident within 12 h after the biological death [94,95,96,97,98]. However, it provides an opportunity to compare ECs from atherosusceptible arteries (e.g., aorta, coronary artery, carotid artery, or femoral artery) and atheroresistant arteries such as the internal thoracic artery or the radial artery. Another challenging task is to compare microvascular ECs from non-diabetic patients and those who suffered from diabetic retinopathy, diabetic nephropathy, or other forms of diabetic microangiopathy. This might uncover the molecular signatures of endothelial dysfunction in diabetes and pave the way for its correction using oral hypoglycemic agents. However, neither HUVEC nor cadaveric ECs can be applied for the selection of personalized therapies from bench to bedside.

An objective investigation of endothelial heterogeneity in vitro is complicated by requirement to isolate distinct EC lineages from the same patient as well as by differences in: (1) isolation efficiency and variable quantities of ECs at the 1st passage; (2) proliferative capacity of isolated ECs and number of doublings or passages before the senescence; (3) number of immunomagnetic separations required to remove fibroblasts for obtaining pure EC culture; (4) rates of cellular senescence which depends on the initial amount of isolated ECs, number of immunomagnetic selections, and doublings required to reach confluence. In general, isolation of HMVEC requires a higher number of immunomagnetic selections (from 5 to 6 in most cases) and passages to obtain a pure culture that HSaVEC or HITAEC due to the relatively small amounts of adipose tissue that can be safely excised during the surgery. Lower rates of successful isolation of HITAEC in comparison with HSaVEC are probably also related to the shorter length and smaller diameter of ITA in comparison with SV conduits. Yet, we were able to conduct a synchronous isolation of HSaVEC, HITAEC, and HMVEC from 6/40 (≈15%) patients, allowing for a comparison of their global gene expression profiles in future studies. To explore the influence of the culture medium on the proliferation and angiogenic activity of HSaVEC, HITAEC, and HMVEC, we compared several commercially available formulations (EndoBoost, growth factor- and serum-enriched EndoBoost Plus from AppScience Products, MesoEndo from Cell Applications, and EGM-2MV from Lonza). All mentioned EC types showed stable proliferation and adequate in vitro angiogenesis, albeit EndoBoost Plus medium was superior for these applications in our experimental setting.

Among the limitations of our approach was the inability to isolate ECs from atherosusceptible artery (e.g., coronary artery or aorta), as they are not excised during CABG surgery. Further perspectives include the extended molecular characterization of HSaVEC, HITAEC, and HMVEC by RNA sequencing and comparison of their responses to endothelial dysfunction triggers such as calciprotein particles, mitomycin C, pro-inflammatory cytokines, palmitic acid, urea, lipopolysaccharide, the SARS-CoV-2 spike protein S1 subunit, endothelial nitric oxide inhibitors, or nutrient deprivation. Profiling of pro-inflammatory cytokines by means of dot blotting, enzyme-linked immunosorbent assay, or multi-analyte detection with fluorescent-labeled microspheres may pinpoint the molecular patterns of pathological activation with regard to distinct EC specifications if combined with RNA sequencing or reverse transcription-quantitative polymerase chain reaction to verify the expression of the corresponding genes. Similar approach can be applied for assessing the release of basement membrane components (e.g., perlecan, nidogen-1, type IV collagen subunits, and laminin) and sub-endothelial extracellular matrix components (e.g., thrombospondin-1, von Willebrand factor, matrix metalloproteinase 2, and tissue inhibitor of metalloproteinases 1 and 2), as ECs release them in considerable amounts both at baseline conditions and following the exposure to endothelial dysfunction triggers. The latter is accompanied by shedding of constitutive (e.g., CD31/platelet endothelial cell adhesion molecule 1, VE-cadherin, CD146/melanoma cell adhesion molecule, CD105/endoglin, and CD309/vascular endothelial growth factor receptor 2/kinase insert domain receptor) and inducible (vascular cell adhesion molecule 1, intercellular cell adhesion molecule 1, and E-selectin) endothelial receptors, some of which have been suggested as sensitive and specific markers of endothelial dysfunction.

Although it remains unclear whether the EC specification defines a differential response to endothelial dysfunction triggers, there are several clinical arguments in favor of this assumption. Despite an epidemiological link between atherosclerotic vascular disease and venous thrombosis [99,100,101,102,103,104], these disorders have slightly different distribution patterns in the population [105,106] as atherosclerosis is intimately associated with dyslipidemia and arterial hypertension [107,108,109], whilst the risk of venous thromboembolism is largely defined by a genetic predisposition and hormonal changes [110,111,112]. Likewise, although the risk of microvascular dysfunction in COVID-19 is increased in patients with atherosclerotic vascular disease or deep vein thrombosis [9,113,114,115,116], it also occurs in patients without cardiovascular disease [117]. These observations correspond to our previous studies, which showed augmented transcription and elevated basal secretion of pro-inflammatory cytokines by arterial ECs in comparison with venous ECs [54,55]. Simulation of paracrine interactions between human coronary artery endothelial cells (HCAEC), HITAEC, and HSaVEC by using a conditioned medium approach identified an increased expression of *CXCL8*, *ICAM1*, and *SELE* genes in HITAEC as compared to HSaVEC [54]. RNA sequencing of HCAEC and HITAEC, which were cultured in serum-free medium under laminar flow or static cell culture conditions, revealed the overexpression of pro-inflammatory genes, including those encoding cytokines and chemokines (in particular interleukin-6), in HITAEC but not HCAEC [55]. Verification of differentially expressed molecular terms in laminar flow culture by reverse transcription-quantitative polymerase chain reaction showed higher expression of *ICAM1*, *SELE*, *IL6*, *CXCL8*, and *CCL2* genes in HITAEC as compared to HCAEC [55]. In concert with these findings, HITAEC exhibited elevated expression of the genes encoding pro-inflammatory cell adhesion molecules (*VCAM1*, *ICAM1*, and *SELE*) and chemokines (*CXCL8* and *CCL2*), under static conditions [55]. In this study, HSaVEC, HITAEC, and HMVEC also showed an overexpression of ICAM1 protein and elevated release of interleukin-6 and MCP-1/CCL2 upon TNF-α stimulation, which has been accompanied by the upregulation of the corresponding genes, together suggesting their relevance for studying endothelial dysfunction. Collectively, this suggests specific risk factors and molecular pathways underlying the pathological activation of HSaVEC, HITAEC, and HMVEC, highlighting the advantages of their isolation from the same patient during CABG surgery.

## 4. Materials and Methods

The study was conducted according to the latest revision of the Declaration of Helsinki (2013), and the study protocol was approved by the Local Ethical Committee of the Research Institute for Complex Issues of Cardiovascular Diseases (Kemerovo, Russia, protocol code 028/03/2024, date of approval: 28 March 2024). Written informed consent has been provided by all study participants after receiving a full explanation of the study’s purposes. In general, isolation of HSaVEC, HITAEC, and HMVEC included the following: (1) collection of SV, ITA, and adipose tissue segments during the CABG surgery; (2) enzymatic dissociation with collagenase type IV for degrading the basement membrane; (3) consecutive filtration through the cell strainers (exclusively for HMVEC); (4) positive immunomagnetic separation using anti-CD31-antibody-coated magnetic beads; (5) EC culture on fibronectin- or gelatin-coated flasks with repeated positive immunomagnetic separations until the formation of EC monolayer; (6) flow cytometry immunophenotyping using anti-CD31/platelet endothelial cell adhesion molecule 1 (PECAM1), anti-CD146, anti-CD34, and anti-CD90 fluorescent-conjugated antibodies; (7) testing of proliferative and angiogenic activity by means of cell viability and proliferation assay, bioimpedance measurements, and tube formation assay. All procedures were performed in the laminar hood under sterile conditions. All steps were performed at room temperature unless otherwise stated.

### 4.1. Reagents and Consumables

Cell counting chamber slide (DS-50, RWD Life Science Co., Ltd., Shenzhen, China; C10228, Thermo Fisher Scientific, Waltham, MA, USA);Petri dish, 100 mm diameter (CCD-100, Wuhan Servicebio Technology Co., Ltd., Wuhan, China);Straight anatomical eye tweezer, 100 mm length (J-16-131, Surgicon, Sialkot, Pakistan);Eye scissors with straight pointed tips, 115 mm length (J-22-211A, Surgicon, Sialkot, Pakistan);Sterile gauze wipes;Cell culture flasks, 25 cm^2^ surface area (T-25, 707003, Wuxi NEST Biotechnology Co., Ltd., Wuxi, China);Cell culture flasks, 75 cm^2^ surface area (T-75, 07-8075, Biologix Plastic (Changzhou) Co., Ltd., Changzhou, China);Falcon tubes, 15 mL volume (601002, Wuxi NEST Biotechnology Co., Ltd., Wuxi, China);Falcon tubes, 50 mL volume (602002, Wuxi NEST Biotechnology Co., Ltd., Wuxi, China);Open-top round-bottom polystyrene flow cytometry tube, 5 mL volume (BIO-U-12-C, Sovtech, Moscow, Russia);Syringe, 5 mL volume with 22G needle (0.7 × 40 mm, SFM Hospital Products, Berlin, Germany);Single-channel adjustable volume pipette, 1–10 mL volume (4640072, Thermo Fisher Scientific, Waltham, MA, USA);Single-channel adjustable volume pipette, 100–1000 μL volume (4640062, Thermo Fisher Scientific, Waltham, MA, USA);Single-channel adjustable volume pipette, 20–200 μL volume (4640052, Thermo Fisher Scientific, Waltham, MA, USA);Single-channel adjustable volume pipette, 1–10 μL volume (4640012, Thermo Fisher Scientific, Waltham, MA, USA);Pipette tips (10 mL, 9402152, Thermo Fisher Scientific, Waltham, MA, USA; 1000 μL, HP2036-4, Yancheng Huida Medical Instruments Co., Ltd., Yancheng, China; 200 μL, HP2032-6, Yancheng Huida Medical Instruments Co., Ltd., Yancheng, China; 10 μL, TP-10P-C-F, Wuhan Servicebio Technology Co., Ltd., Wuhan, China);Serum-supplemented EC culture medium (EndoBoost, EB1, AppScience Products, Moscow, Russia; EGM-2MV, Lonza Bioscience, Basel, Switzerland; MesoEndo Growth, Cell Applications, San Diego, CA, USA);Growth factor- and fetal bovine serum-enriched EC culture medium (EndoBoost Plus, EB2, AppScience Products, Moscow, Russia);Serum-free EC culture medium (EndoLife, EL1, AppScience Products, Moscow, Russia);Cell strainers for 50 mL Falcon tubes, 100 μm pore diameter (15-1100, Biologix Plastic (Changzhou) Co., Ltd., Changzhou, China);Cell strainers for 50 mL Falcon tubes, 70 μm pore diameter (15-1070, Biologix Plastic (Changzhou) Co., Ltd., Changzhou, China);Cell strainers for 50 mL Falcon tubes, 40 μm pore diameter (15-1040, Biologix Plastic (Changzhou) Co., Ltd., Changzhou, China);Fibronectin from bovine plasma (1.4.11, Biolot, St. Petersburg, Russia);0,1% gelatin solution (1.4.6, Biolot, St. Petersburg, Russia);Collagenase type IV from *Clostridium histolyticum* (≥160 U/mg, GC305015–100 mg, Wuhan Servicebio Technology Co., Ltd., Wuhan, China);Dissociation buffer (0.15% collagenase type IV from *Clostridium histolyticum*, dissolved in serum-free EndoLife medium);Anti-CD31-antibody-coated magnetic beads (Dynabeads CD31, 11155D, Thermo Fisher Scientific, Waltham, MA, USA);Dulbecco’s Phosphate-Buffered Saline without calcium and magnesium (1 × DPBS, 1.2.4.7., Biolot, St. Petersburg, Russia);0.25% trypsin-ethylenediaminetetraacetic acid (EDTA) in Hank′s Balanced Salt Solution (HBSS) (P043p, PanEco, Moscow, Russia);Fetal bovine serum (FBS, 1.1.6.1, Biolot, St. Petersburg, Russia);Trypsin neutralizing solution (5% FBS dissolved in 1 × DPBS, Biolot, St. Petersburg, Russia);Magnetic beads wash buffer (0.1% FBS dissolved in 1 × DPBS, Biolot, St. Petersburg, Russia);Deionized water (P009p, PanEco, Moscow, Russia);Sodium azide (S2002, Sigma-Aldrich, Saint Louis, MO, USA);Phycoerythrin-Cyanine 7 (PC7)-conjugated anti-CD146 antibody (mouse monoclonal IgG1, 2405040, Sony Biotechnology, San Jose, CA, USA);Fluorescein isothiocyanate (FITC)-conjugated anti-CD31 antibody (mouse monoclonal IgG1, 2115520, Sony Biotechnology, San Jose, CA, USA);Alexa Fluor 700 (AF700)-conjugated anti-CD90 antibody (mouse monoclonal IgG1, 2240600, Sony Biotechnology, San Jose, CA, USA);Pacific Blue (PB)-conjugated anti-CD34 antibody (mouse monoclonal IgG1, 2317560, Sony Biotechnology, San Jose, CA, USA);PC7-conjugated isotype control antibody (mouse monoclonal IgG1, 2600630, Sony Biotechnology, San Jose, CA, USA);FITC-conjugated isotype control antibody (mouse monoclonal IgG1, 2600540, Sony Biotechnology, San Jose, CA, USA);AF700-conjugated isotype control antibody (mouse monoclonal IgG1, 2600720, Sony Biotechnology, San Jose, CA, USA);PB-conjugated isotype control antibody (mouse monoclonal IgG1, 2600755, Sony Biotechnology, San Jose, CA, USA);Chambered polymer coverslip with individual wells (µ-Slide 8-well, 80826, Ibidi, Grafelfing, Germany);4% paraformaldehyde solution (158127, Sigma-Aldrich, St. Louis, MO, USA);Triton X-100 (SYS-Q0011-1.0, Suzhou Yacoo Science Co., Ltd., Suzhou, China);1% bovine serum albumin solution (P091E, PanEco, Moscow, Russia);Unconjugated anti-CD31/PECAM1 antibody (mouse monoclonal IgG1, ab9498, Abcam, Cambridge, UK);Unconjugated anti-von Willebrand factor antibody (rabbit monoclonal IgG, A21054, ABclonal Biotechnology Co., Ltd., Wuhan, China);Donkey anti-mouse pre-adsorbed Alexa Fluor 555-conjugated antibody (ab150110, Abcam, Cambridge, UK);Donkey anti-rabbit pre-adsorbed Alexa Fluor 488-conjugated antibody (ab150061, Abcam, Cambridge, UK);4′,6-diamidino-2-phenylindole (DAPI, 10 µg/mL, 2996, Lumiprobe, Moscow, Russia);ProLong Gold Antifade (P36934, Thermo Fisher Scientific, Waltham, MA, USA);96-well tissue culture-treated cell culture plates (703001, Wuxi NEST Biotechnology Co., Ltd., Wuxi, China);Cell Cytotoxicity Assay Kit (Colorimetric) (ab112118, Abcam, Cambridge, UK);16-well E-Plates for the xCELLigence Real-Time Cell Analysis Dual Purpose Instrument (300600890, Agilent Technologies, Santa Clara, CA, USA);24-well tissue culture-treated cell culture plates (702001, Wuxi NEST Biotechnology Co., Ltd., Wuxi, China);Standard Star Matrigengel (082704, Xiamen Mogengel Biotechnology Co., Ltd., Xiamen, China);Eukaryotic bioactive TNF-α (PSG250-50, Sci-Store, Moscow, Russia);TRIzol (15596018, Thermo Fisher Scientific, Waltham, MA, USA);Radioimmunoprecipitation lysis and extraction buffer (89901, Thermo Fisher Scientific, Waltham, MA, USA);Protease and phosphatase inhibitor cocktail (Halt, 78444, Thermo Fisher Scientific, Waltham, MA, USA);Bicinchoninic acid protein assay Kit (23227, Thermo Fisher Scientific, Waltham, MA, USA);cDNA synthesis kit (M-MuLV–RH First Strand cDNA Synthesis Kit, R01-250, Evrogen, Moscow, Russia);Reverse transcriptase (M-MuLV–RH, R03-50, Evrogen, Moscow, Russia);Customized primers (Evrogen, Moscow, Russia);qPCR master mix (BioMaster HS-qPCR Lo-ROX SYBR Master Mix, MHR031-2040, Biolabmix, Novosibirsk, Russia);Molecular weight marker (Chameleon Duo Pre-Stained Protein Ladder, 928–60,000, LI-COR Biosciences, Lincoln, NE, USA);Sample buffer for the protein denaturation (OrangeMark, K-023, Molecular Wings, Kemerovo, Russia);Polyacrylamide gels for the electrophoretic protein separation (NuPAGE 4–12% Bis-Tris protein gel, NP0335BOX, Thermo Fisher Scientific, Waltham, MA, USA);Running buffer for the electrophoretic protein separation (G-RUN MES, K-021, Molecular Wings, Kemerovo, Russia);Antioxidant, the electrophoretic protein separation (G-NOOOX, K-027, Molecular Wings, Kemerovo, Russia)Nitrocellulose protein transfer stacks (IB23001, Thermo Fisher Scientific, Waltham, MA, USA);Protein-free blocking solution (Block’n’Boost!, K-028, Molecular Wings, Kemerovo, Russia);iBind Flex Cards (SLF2010, Thermo Fisher Scientific, Waltham, MA, USA);Unconjugated anti-ICAM1 antibodies (rabbit monoclonal IgG, A19300, ABclonal Biotechnology Co., Ltd., Wuhan, China; rabbit polyclonal IgG, AF6088, Affinity Biosciences, Cincinnati, OH, USA);Unconjugated anti-β-actin antibody (mouse monoclonal IgG, SLM-33036M, Sunlong Biotech, Hangzhou, China);IRDye 800CW-conjugated goat anti-rabbit antibody (926-32211, LI-COR Biosciences, Lincoln, NE, USA);IRDye 680RD-conjugated goat anti-mouse antibody (926-68070, LI-COR Biosciences, Lincoln, NE, USA);Acid Black 1 (A2097, Tokyo Chemical Industry, Tokyo, Japan);ELISA kit for measuring human interleukin-6 (A-8768, Vector-Best, Koltsovo, Russia);ELISA kit for measuring human MCP-1/CCL2 (A-8782, Vector-Best, Koltsovo, Russia).

### 4.2. Equipment

Tissue culture hood (BMB-II-Laminar-S-1.8-NEOTERIC, Laminar Systems, Miass, Russia);Microcentrifuge vortex (FV-2400, Biosan, Riga, Latvia);EasySep cell separation magnet (18000, STEMCELL Technologies, Vancouver, BC, Canada);Benchtop multi-purpose centrifuges (5804R, Eppendorf, Hamburg, Germany; Microfuge 20R, Beckman Coulter, Brea, CA, USA);CO_2_ incubator (MCO-18AIC, Sanyo, Panasonic Corporation, Osaka, Japan);Automated cell counter (Countess II, Thermo Fisher Scientific, Waltham, MA, USA);Orbital shaker (OS-200, Biosan, Riga, Latvia);Flow cytometer (CytoFLEX, Beckman Coulter, Brea, CA, USA);Inverted fluorescence microscope (AxioObserver.Z1, Carl Zeiss, Jena, Germany);Laser scanning confocal microscope (LSM 700, Carl Zeiss, Jena, Germany);Ultraviolet-visible microplate spectrophotometer (Multiskan Sky, Thermo Fisher Scientific, Waltham, MA, USA);xCELLigence Real-Time Cell Analysis Dual Purpose (RTCA DP) instrument (Agilent Technologies, Santa Clara, CA, USA);Microvolume ultraviolet-visible spectrophotometer (NanoDrop 2000, Thermo Fisher Scientific, Waltham, MA, USA);96-well thermal cycler (Veriti, Thermo Fisher Scientific, Waltham, MA, USA);Real-time PCR system (ViiA 7, Thermo Fisher Scientific, Waltham, MA, USA);Vertical mini-protein gel electrophoresis system (XCell SureLock Mini-Cell, EI0001, Thermo Fisher Scientific, Waltham, MA, USA);Dry blotting system (iBlot 2 Gel Transfer Device, Thermo Fisher Scientific, Waltham, MA, USA);Automated Western blot processing system (iBind Flex Western Device, Thermo Fisher Scientific, Waltham, MA, USA);Fluorescent Western blot imaging system (Odyssey XF, LI-COR Biosciences, Lincoln, NE, USA);Digital microplate shaker (Titramax 1000, Heidolph, Schwabach, Germany);Microplate washer (Wellwash Versa, Thermo Fisher Scientific, Waltham, MA, USA);Microplate spectrophotometer (Multiskan Sky, Thermo Fisher Scientific, Waltham, MA, USA).

### 4.3. Coating of Cell Culture Flasks with Fibronectin or Gelatin

A stock solution of fibronectin was prepared by dissolving 1 mg lyophilized fibronectin in 1 mL DPBS (1 mg/mL) and incubating at 37 °C for 30 min for complete dissolution. A working solution was prepared by diluting the stock solution 1:100 in DPBS. Cell culture flasks were coated with fibronectin (10 μg/mL, 1 μg per 1 cm^2^ surface area or 1 mL per 10 cm^2^ surface area, 25 μg in 2.5 mL per T-25 flask or 75 μg in 7.5 mL per T-75 flask) or 0.1% gelatin (1 mL per 10 cm^2^ surface area, 2.5 mL per T-25 flask or 7.5 mL per T-75 flask). Immediately after the coating, we made a visual confirmation that fibronectin or gelatin had evenly coated the surface area. Fibronectin-coated flasks were incubated at room temperature for 1 h. Gelatin-coated flasks were incubated either at 37 °C for 30 min or at 4 °C for 12 h. After the incubation, we removed the fibronectin or gelatin solution and left the cell culture flasks in the laminar hood at room temperature for 30 min to dry the protein coating. Coated cell culture flasks were either used immediately or stored at 4 °C for 1 or 2 weeks. Before use, the flasks were equilibrated to room temperature.

### 4.4. Isolation of HSaVEC and HITAEC

Excised segments of SV (≈2 cm length) and ITA (≈1 cm length) were immediately placed into the sterile 50 mL Falcon tube pre-filled with ice-cold (4 °C) 5 mL EndoBoost medium and stored at 4 °C until the beginning of the isolation. The time frame from tissue excision to HSaVEC/HITAEC isolation did not exceed 5 h. All working solutions were pre-warmed to 37 °C until the beginning of the isolation.SVs and ITAs were placed into the Petri dish with DPBS (pre-warmed to 37 °C), washed to remove the blood, and transferred into another Petri dish with pre-warmed DPBS.Using a tweezer and the scissors, we performed a longitudinal section along the entire SV or ITA and washed the blood vessel of any remaining blood if needed.Sectioned SVs and ITAs were placed into separate 15 mL Falcon tubes, filled with dissociation buffer (0.15% collagenase type IV in serum-free EndoLife medium) in 1:1 (SV/ITA to dissociation buffer) ratio and incubated at 37 °C for 60 min, with a brief and mild vortexing each 10 min (6 times per incubation) to facilitate EC detachment. After incubation, SVs and ITAs were gently vortexed for 2 min to promote EC detachment and irrigated with the trypsin-neutralizing solution.Then, SV and ITA segments were removed from the tubes and were rinsed with trypsin neutralizing solution while being held in the air. The flush was collected into the same 15 mL Falcon tube.Detached ECs were centrifuged at 220× *g* for 5 min at room temperature. The supernatant was discarded, and the cell pellet was resuspended in 1 mL EndoBoost medium, with the subsequent quantification using an automated cell counter.After the counting, ECs were resuspended in the EndoBoost medium to a 1.5 × 10^5^ cells per 1 mL concentration, seeded into the T-25 fibronectin-coated cell culture flasks (5 mL medium containing 0.75 × 10^6^ cells per flask), and placed into the CO_2_ incubator (37 °C, 5% CO_2_, high humidity).If the isolation was successful, HSaVEC and HITAEC colonies (and also single ECs) were visualized within 24 h, and the medium was changed to EndoBoost Plus to accelerate cell growth. HSaVEC/HITAEC cultures were visually inspected daily, and the medium was changed thrice a week. On day 3 or 4, we started to observe elongated, spindle-shaped, or stellate fibroblast-like cells, which were removed at day 6 by the positive selection of the ECs using magnetic beads coated with anti-CD31 antibodies (25 μL beads per ≤1 × 10^8^ cells/mL). Immunomagnetic separation-assisted enrichment of the ECs was repeated until the formation of an EC monolayer. The isolation of HSaVEC and HITAEC is represented in Figure 11.

### 4.5. Isolation of HMVEC from Subcutaneous Adipose Tissue

Excised segments of adipose tissue (≈5 cm^3^) were immediately placed into the sterile 50 mL Falcon tube pre-filled with ice-cold (4 °C) 5 mL EndoBoost medium and stored at 4 °C until the beginning of the isolation. The time frame from tissue excision to the HMVEC isolation did not exceed 5 h. All working solutions were pre-warmed to 37 °C until the beginning of the isolation.Adipose tissue fragments were placed into the Petri dish with DPBS (pre-warmed to 37 °C), washed to remove the blood, and transferred into another Petri dish with pre-warmed DPBS.Using a tweezer and the scissors, adipose tissue (from 1.0 to 1.5 cm^3^) was separated from the adjacent stroma (3.5–4.0 cm^3^) and minced into small pieces. Chopped adipose tissue was placed into 50 mL Falcon tubes, filled with dissociation buffer (0.15% collagenase type IV in serum-free EndoLife medium) in 2:1 (adipose tissue: dissociation buffer) ratio and incubated at 37 °C for 60 min, with a brief and mild vortexing each 10 min (6 times per incubation) to facilitate EC detachment. After the incubation, adipose tissue segments were gently vortexed for 2 min to promote EC detachment, and the tube was filled with trypsin neutralizing solution up to 50 mL.Adipose tissue was consecutively removed by filtering the lysate through the cell strainers with 100, 70, and 40 μm pore diameters. For each filtration step, we used a fresh 50 mL Falcon tube.The cell suspension was centrifuged at 220× *g* for 10 min at room temperature. The supernatant was discarded, and the cell pellet was resuspended in 1 mL EndoBoost medium, with the subsequent quantification using the automated cell counter.Then, a positive immunomagnetic separation was performed by adding 1 mL cell suspension to anti-CD31-antibody-coated magnetic beads (25 μL beads per ≤1 × 10^8^ cells/mL). After the last washing, bead-bound ECs were resuspended in 1 mL of EndoBoost medium, seeded into T-25 fibronectin-coated cell culture flasks pre-filled with 5 mL EndoBoost medium, and placed in a CO_2_ incubator (37 °C, 5% CO_2_, high humidity).If the isolation was successful, HMVEC colonies (and also single ECs) were visualized within 24 h, and the medium was changed to EndoBoost Plus to accelerate the cell growth. HMVEC cultures were visually inspected daily, and the medium was changed thrice a week. On day 3 or 4, we started to observe elongated, spindle-shaped, or stellate fibroblast-like cells, which were removed at day 6 by the repeated positive selection of the ECs using magnetic beads coated with anti-CD31 antibodies. Magnetic separation-assisted enrichment of the ECs was repeated again until the formation of an EC monolayer. The isolation of adipose tissue-derived HMVEC is represented in Figure 12.

### 4.6. Positive Immunomagnetic Separation of HSaVEC, HITAEC, and HMVEC

Following 3 min vortexing, 25 μL anti-CD31-antibody-coated magnetic beads (sufficient for the treatment of 1 × 10^8^ cells/mL) were resuspended in 1 mL wash buffer in the open-top round-bottom flow cytometry tube, which was then positioned into the EasySep magnet for 1 min to sediment washed magnetic beads. The supernatant was discarded while still holding the tube within the magnet to retain the magnetic beads. The tube was then withdrawn from the magnet, and the washed magnetic beads were resuspended in 25 μL wash buffer.After the removal of cell culture medium from T-25 flasks, ECs were washed with 5 mL DPBS and incubated in 1 mL 0.25% trypsin-EDTA at 37 °C for 5 min to detach the cells. If the ECs were poorly detached from the surface, the flasks were gently tapped on the bench to stimulate the detachment. Cell dissociation was controlled by phase contrast microscopy.After the cell detachment, trypsin neutralizing solution (3 mL) was added to the flask in order to neutralize 1 mL of 0.25% trypsin-EDTA, and 4 mL of EC suspension were transferred into the 15 mL Falcon tube. The flask was flushed with 2 mL trypsin neutralizing solution into the same tube to collect the residual ECs. The flask was then examined by phase contrast microscopy to ensure complete cell detachment.EC suspension (6 mL) was centrifuged at 220× *g* for 5 min at room temperature. The supernatant was discarded, and the cell pellet was resuspended in 1 mL of trypsin-neutralizing solution. The cell suspension was then quantified using an automated cell counter and transferred into an open-top round-bottom flow cytometry tube.Next, 1 mL EC suspension was added to anti-CD31-antibody-coated magnetic beads (25 μL beads per ≤1 × 10^8^ cells/mL) from Step 1, gently vortexed for 10 s, and incubated on the orbital shaker at a low speed for 20 min at 4 °C.After the incubation, the flow cytometry tube with a cell suspension was positioned into the EasySep magnet for 2 min at room temperature. The supernatant was discarded while still holding the tube within the magnet so as to retain ECs conjugated with the magnetic beads. After removing the tube from the magnet, ECs bound to the magnetic beads were resuspended in 1 mL trypsin neutralizing solution and vortexed at low speed for 3 s.Step 6 has been repeated twice (3 times in total). After the final washing, bead-bound ECs were resuspended in 1 mL of EndoBoost Plus medium and transferred into gelatin-coated T-25 flasks with 5 mL of EndoBoost medium.The next day, the medium was changed to EndoBoost Plus to accelerate the cell growth.

### 4.7. Flow Cytometry

For the immunophenotyping of the ECs by flow cytometry, we performed a dissociation of the ECs from the gelatin by 0.25% trypsin-EDTA, washed ECs with 0.1% sodium azide (dissolved in 1 × DPBS) to prevent non-specific cell activation, resuspended, and selected 100 μL cell suspension containing 1 × 10^5^ cells. The combination of mouse monoclonal PC7-conjugated anti-CD146, FITC-conjugated anti-CD31, AF700-conjugated anti-CD90, and PB-conjugated anti-CD34 antibodies was used to characterize HSaVEC, HITAEC, and HMVEC phenotype. Isotype control mouse monoclonal PC7-conjugated, FITC-conjugated, AF700-conjugated, and PB-conjugated anti-mouse IgG antibodies were applied for the adjustment for non-specific binding and the respective autofluorescence. EC suspensions were incubated with the antibodies for 30 min in the dark and then analyzed using CytoFlex flow cytometer and CytExpert software (v. 2.4, Beckman Coulter, Brea, CA, USA). The fluorescence intensity from isotype control antibodies was set as a signal threshold. The compensation was calculated using the Fluorescence Minus One approach, where the cell suspension is stained with all fluorescent-conjugated antibodies except one. Fluorescence minus one procedure was conducted for each of the fluorophores. To exclude debris and doublets, we demarcated the target gate on a forward scatter height/forward scatter area plot and then transferred this gate onto a forward scatter/side scatter plot, which shows cell size and granularity (internal complexity), and where we selected the final gate employed to analyze all samples. Following the collection of ≥5000 target events, we evaluated the proportion of CD146^+^CD31^+^CD34^+^CD90^−^ cells within each sample (≥99% was considered as a pure EC culture).

### 4.8. Confocal Microscopy

HSaVEC, HITAEC, and HMVEC were seeded into the 8-well cell culture chambers for 3 h to ensure a proper adhesion to the polymer coverslip, fixed with 4% paraformaldehyde for 10 min, permeabilized in Triton X-100 for 15 min, and blocked in 1% bovine serum albumin for 1 h to prevent non-specific binding. Then, ECs were stained with mouse anti-CD31/PECAM1 (1:500 dilution, ab9498, Abcam, Cambridge, UK) and rabbit anti-von Willebrand factor (1:500, ABclonal Biotechnology Co., Ltd., Wuhan, China) primary antibodies for 16 h at 4 °C. The next day, cells were stained with donkey anti-mouse pre-adsorbed Alexa Fluor 555-conjugated (1:500, ab150110, Abcam, Cambridge, UK) and donkey anti-rabbit pre-adsorbed Alexa Fluor 488-conjugated (1:500, ab150061, Abcam, Cambridge, UK) secondary antibodies for 1 h at room temperature. Counterstaining was performed with 4′,6-diamidino-2-phenylindole (DAPI, 10 µg/mL) for 30 min. At all stages, washing was conducted with DPBS. Coverslips were mounted with ProLong Gold Antifade. Visualization was performed by confocal microscopy.

### 4.9. Functional Assays

To assess cell proliferation and viability, HSaVEC, HITAEC, and HMVEC were seeded into 96-well gelatin-coated plates (10,000 cells per well) in the EndoBoost, EndoBoost Plus, MesoEndo, or EGM-2MV medium (200 μL per well). The next day, we changed the medium (180 μL per well), added 20 μL of the reagent (ab112118, Abcam, Cambridge, UK), and incubated ECs in the CO_2_ incubator (37 °C, 95% air: 5% CO_2_, and high humidity) with the consecutive spectrophotometric measurements of the optical density at 570 (OD_570_) and 605 (OD_605_) nm wavelength at 24, 48, and 72 h time points. The higher OD_570_ to OD_605_ ratio indicated the higher proliferation and viability of the ECs. Adjustment for the background signal was conducted by the subtraction of OD_570_ to OD_605_ ratio in the control wells without the ECs from the obtained ratios in all wells with the ECs.

The proliferation of HSaVEC, HITAEC, and HMVEC was further evaluated using the xCELLigence Real-Time Cell Analysis Dual Purpose instrument for noninvasive electrical impedance monitoring. ECs in the EndoBoost Plus medium were seeded in 16-well E-plates (2 × 10^4^ cells per well) with subsequent measurements of impedance over 72 h. Proliferation capability was defined as cell index doubling time, which was calculated automatically in the Real-Time Cell Analysis Software (v. 2.0, Agilent Technologies, Santa Clara, CA, USA).

Angiogenic activity of HSaVEC, HITAEC, and HMVEC was assessed by adding ECs in the EndoBoost, EndoBoost Plus, MesoEndo, or EGM-2MV medium into the 24-well plates (1 × 10^5^ cells per well) pre-filled (30 min at room temperature to achieve polymerization) with Standard Star Matrigengel (200 μL per well). After 24 h of incubation, the formation of capillary-like tubes was assessed by phase contrast microscopy.

### 4.10. TNF-α Activation Assay

To investigate the molecular profile in response to pro-inflammatory stimulation, HSaVEC, HITAEC, and HMVEC were treated with DPBS or eukaryotic bioactive TNF-α (20 ng/mL) in a serum-free EC culture medium for 24 h. Upon the withdrawal of medium and washing in ice-cold (4 °C) DPBS, ECs were lysed with either TRIzol to isolate RNA (*n* = 4 patients per EC type, *n* = 12 patients per group) or radioimmunoprecipitation lysis and extraction buffer supplemented with protease and phosphatase inhibitors to extract total protein, using a microcentrifuge and according to the respective manufacturers’ protocols. RNA and protein quantification were conducted using a microvolume ultraviolet-visible spectrophotometer or bicinchoninic acid assay and a microplate spectrophotometer, respectively, in accordance with the manufacturer’s protocols. The cell culture medium was centrifuged at 2000× *g* to sediment cell debris and then frozen at −80 °C until used for ELISA.

### 4.11. Gene Expression Analysis

Gene expression analysis in HSaVEC, HITAEC, and HMVEC was performed using RT-qPCR as described in our previous studies [54,55,118,119]. Briefly, M-MuLV–RH First Strand cDNA Synthesis Kit and reverse transcriptase M-MuLV–RH were used for the reverse transcription on a 96-well thermal cycler according to the manufacturer’s protocols, and RT-qPCR was carried out employing customized primers (500 nmol/L each, Table 4), cDNA (20 ng), and BioMaster HS-qPCR Lo-ROX SYBR Master Mix on a real-time PCR system according to the manufacturer’s protocol. Quantification of mRNA levels (*ICAM1*, *IL6*, *CCL2*, and *PECAM1* genes) was performed by calculation of ΔCt and by using the 2^−ΔΔCt^ method. Relative transcript levels were expressed as a value relative to the average of the *PECAM1* gene and to the mock (DPBS)-treated group (2^−ΔΔCt^).

### 4.12. Western Blotting

Fluorescent Western blotting was performed as described in our previous studies [55]. Briefly, equal amounts of protein (15 μg per sample) were mixed with a sample buffer in a 6:1 ratio, denatured at 99 °C for 5 min, and then loaded on a 10-well 1.5 mm 4–12% Bis-Tris protein gel. The Chameleon Duo Pre-Stained Protein Ladder was used as a molecular weight marker. Protein separation and transfer were conducted by sodium dodecyl sulfate–polyacrylamide gel electrophoresis at 150 V for 2 h using a running buffer, an antioxidant, and a vertical mini-protein gel electrophoresis system. Protein transfer was performed using nitrocellulose transfer stacks and a dry blotting system as previously described [55]. Blocking of non-specific binding was carried out by incubation of nitrocellulose membranes in protein-free blocking solution for 1 h. The blots were probed with rabbit antibodies to ICAM1 (1:400 dilution) and mouse antibodies to β-actin (loading control, 1:1000 dilution). IRDye 800CW-conjugated goat anti-rabbit and IRDye 680RD-conjugated goat anti-mouse secondary antibodies were used at a 1:1000 dilution. Incubation with the antibodies was performed using a protein-free blocking solution, iBind Flex Cards, and an iBind Flex Western Device according to the manufacturer’s protocols. Fluorescent detection was performed using an Odyssey XF imaging system at 700 nm (685 nm excitation and 730 nm emission) and 800 nm channels (785 nm excitation and 830 nm emission). Total protein staining was performed by incubation of the membranes in 0.1% Acid Black 1 solution for 10 min, followed by 10 min washing in double-distilled water.

### 4.13. Enzyme-Linked Immunosorbent Assay

The levels of interleukin-6 and MCP-1/CCL2 in the pre-centrifuged, serum-free cell culture supernatant (2000× *g*) from DPBS- or TNF-α-treated HSaVEC, HITAEC, and HMVEC were measured via ELISA (*n* = 4 patients per EC type, *n* = 12 patients per group) using the respective kits, a digital microplate shaker, and a microplate washer according to the manufacturer’s protocols. Colorimetric analysis was conducted using a microplate spectrophotometer at a 450 nm wavelength.

### 4.14. Statistical Analysis

Statistical analysis was performed using GraphPad Prism 8 (GraphPad Software, San Diego, CA, USA). For RT-qPCR analysis, data are presented as an arithmetic mean and standard deviation, and groups were compared by a ratio paired *t*-test (i.e., ratios of paired values but not absolute differences between paired values were compared) because of considerable between-sample variability in absolute values and selected metric (fold change). For ELISA measurements, data are presented as median, 25th and 75th percentiles, and range. Groups were compared using the Wilcoxon matched-pairs signed-rank test due to considerable between-sample variability in absolute values. *p*-values ≤ 0.05 were regarded as statistically significant.

## 5. Conclusions

Here, we developed a protocol for isolating HSaVEC, HITAEC, and HMVEC from patients undergoing CABG surgery. Enzymatic degradation of the basement membrane by collagenase type IV, consecutive filtration of cell suspension through the cell strainers (100, 70, and 40 µm pore diameter), and magnetic separation with anti-CD31-antibody-coated magnetic beads collectively ensure relatively high efficiency of the isolation (≈60% for HSaVEC, ≈50% for HITAEC, and ≈20% for HMVEC) and high purity (≥99%) of isolated ECs with high proliferative and angiogenic activity.

## Figures and Tables

**Figure 1 ijms-26-09217-f001:**
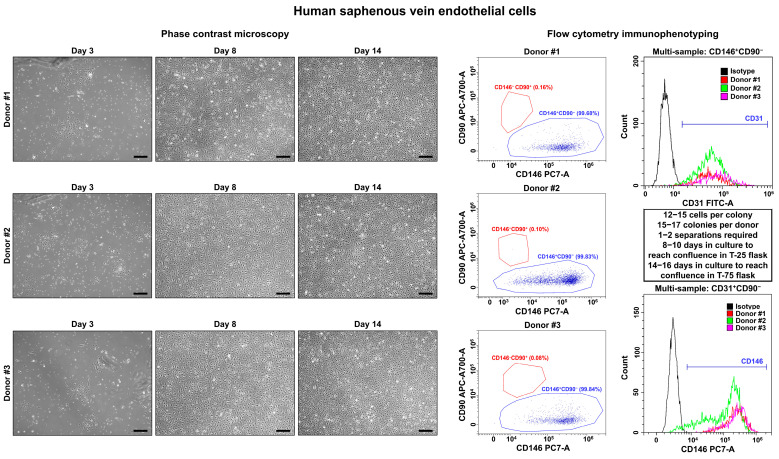
Visualization (phase contrast microscopy, (**left**)) and immunophenotyping (flow cytometry, (**right**)) of primary human saphenous vein endothelial cells (HSaVEC). Three representative donors. Note the single cells or single colonies of HSaVEC on day 3, a cobblestone monolayer in T-25 flasks on day 8, and a cobblestone monolayer in T-75 flasks on day 14. A dot plot (where each dot represents a single cell) indicates the distribution of CD146^+^CD90^−^ (blue color) and CD146^−^CD90^+^ (red color) events, which correspond to endothelial and fibroblast immunophenotype, respectively. Histograms demonstrate the signal intensity from the respective fluorescent-labeled antibody on the *x*-axis and the number of the corresponding events on the *y*-axis. The isotype control is colored black, whilst the samples are colored red, green, and violet. Magnification: ×200. Scale bar = 100 µm.

**Figure 2 ijms-26-09217-f002:**
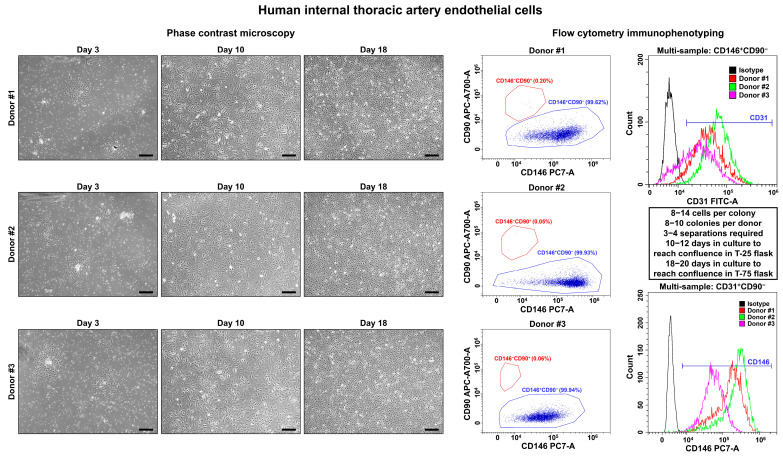
Visualization (phase contrast microscopy, (**left**)) and immunophenotyping (flow cytometry, (**right**)) of primary human internal thoracic artery endothelial cells (HITAEC). Three representative donors. Note the single cells or single colonies of HITAEC on day 3, a cobblestone monolayer in T-25 flasks on day 10, and a cobblestone monolayer in T-75 flasks on day 18. Dot plot (where each dot represents a single cell) indicates the distribution of CD146^+^CD90^−^ (blue color) and CD146^−^CD90^+^ (red color) events, which correspond to endothelial and fibroblast immunophenotype, respectively. Histograms demonstrate the signal intensity from the respective fluorescent-labeled antibody on the *x*-axis and the number of the corresponding events on the *y*-axis. Isotype control is colored black, whilst the samples are colored red, green, and violet. Magnification: ×200. Scale bar = 100 µm.

**Figure 3 ijms-26-09217-f003:**
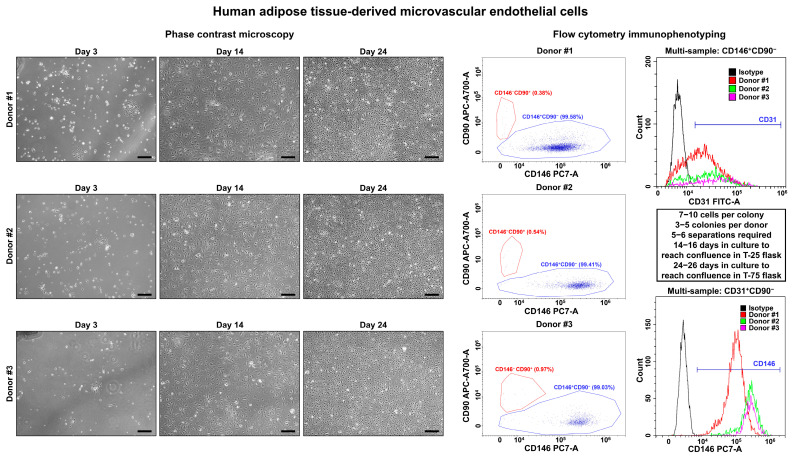
Visualization (phase contrast microscopy, (**left**)) and immunophenotyping (flow cytometry, (**right**)) of primary human adipose tissue-derived microvascular endothelial cells (HMVEC). Three representative donors. Note the single cells or single colonies of HMVEC on day 1, a cobblestone monolayer in T-25 flasks on day 14, and a cobblestone monolayer in T-75 flasks on day 24. Dot plot (where each dot represents a single cell) indicates the distribution of CD146^+^CD90^−^ (blue color) and CD146^−^CD90^+^ (red color) events, which correspond to endothelial and fibroblast immunophenotype, respectively. Histograms demonstrate the signal intensity from the respective fluorescent-labeled antibody on the *x*-axis and the number of the corresponding events on the *y*-axis. Isotype control is colored black, whilst the samples are colored red, green, and violet. Magnification: ×200. Scale bar = 100 µm.

**Figure 4 ijms-26-09217-f004:**
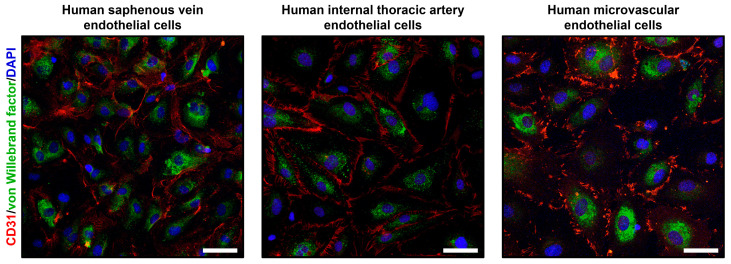
Confocal microscopy of primary human saphenous vein endothelial cells (HSaVEC), human internal thoracic artery endothelial cells (HITAEC), and human adipose tissue-derived microvascular endothelial cells (HMVEC) stained with the antibodies to cluster of differentiation 31/platelet endothelial cell adhesion molecule 1 (CD31/PECAM1, red color) and von Willebrand factor (green color). Nuclei are counterstained with 4′,6-diamidino-2-phenylindole (DAPI, blue color). Magnification: ×200. Scale bar = 50 µm.

**Figure 5 ijms-26-09217-f005:**
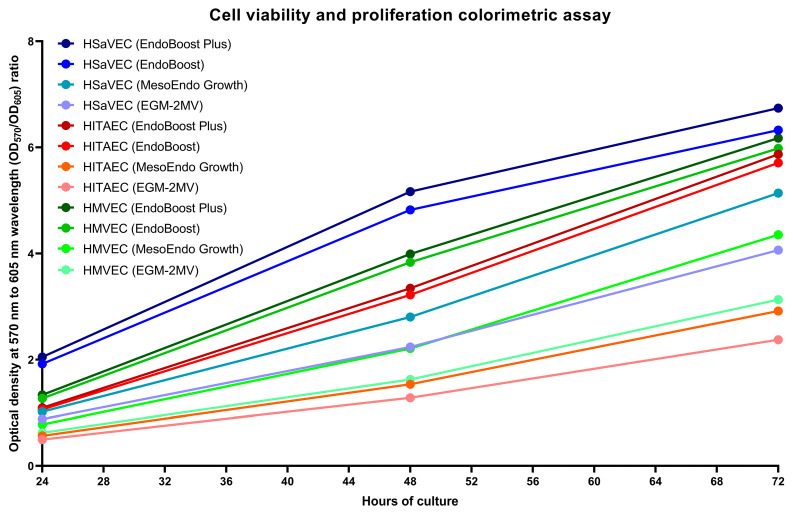
Colorimetric assay of cell viability and proliferation of primary human saphenous vein endothelial cells (HSaVEC), human internal thoracic artery endothelial cells (HITAEC), and human adipose tissue-derived microvascular endothelial cells (HMVEC) cultured in EndoBoost Plus, EndoBoost, MesoEndo Growth, or EGM-2MV medium in 96-well cell culture plates at 24, 48, and 72 h time points. HSaVEC, EndoBoost Plus: navy blue; HSaVEC, EndoBoost: azure blue; HSaVEC, MesoEndo Growth: cyan; HSaVEC, EGM-2MV: light blue; HITAEC, EndoBoost Plus: burgundy; HITAEC, EndoBoost: red; HITAEC, MesoEndo Growth: coral red; HITAEC, EGM-2MV: pink; HMVEC, EndoBoost Plus: dark green; HMVEC, EndoBoost: forest green; HMVEC, MesoEndo Growth: green; HMVEC, EGM-2MV: light green. A higher ratio of optical density at 570 and 605 nm wavelength (OD_570_/OD_605_) indicates higher metabolic intensity registered within the wells.

**Figure 6 ijms-26-09217-f006:**
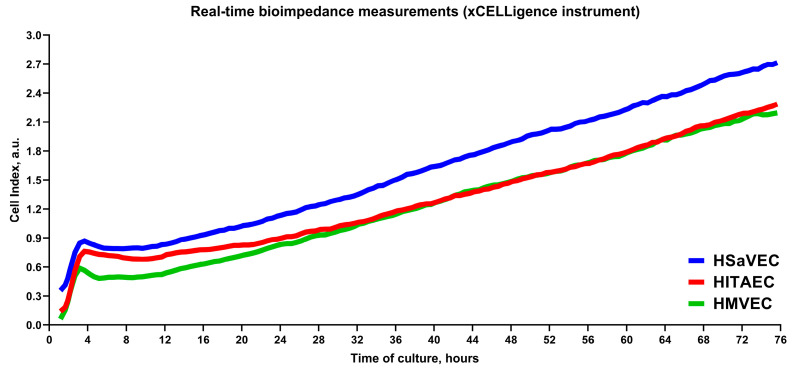
Measurements of bioimpedance indicate the proliferation of primary human saphenous vein endothelial cells (HSaVEC), human internal thoracic artery endothelial cells (HITAEC), and human adipose tissue-derived microvascular endothelial cells (HMVEC) over 72 h. HSaVEC, HITAEC, and HMVEC impedance values (reflected by cell index) are indicated by blue, red, and green lines, respectively.

**Figure 7 ijms-26-09217-f007:**
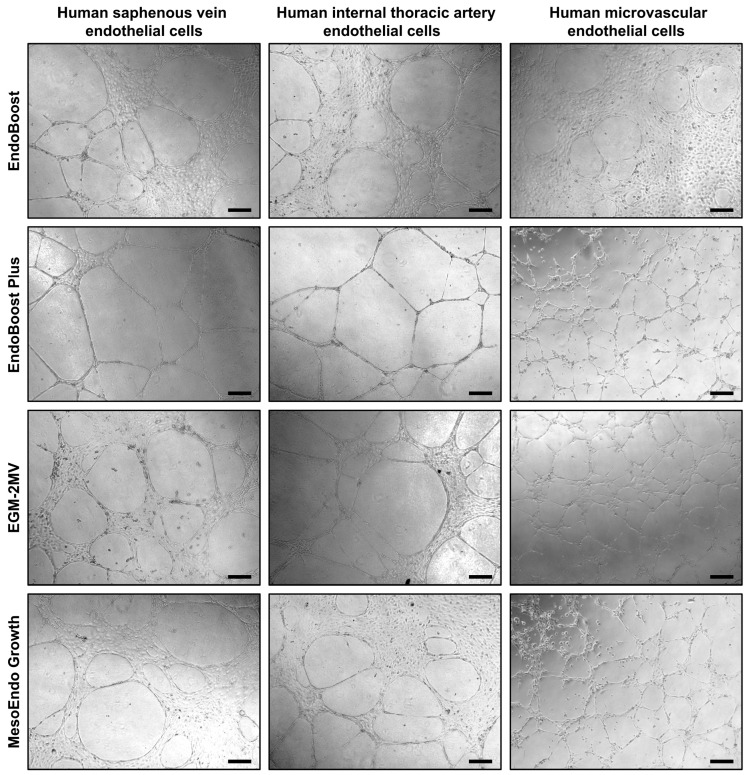
Formation of capillary-like tubes by primary human saphenous vein endothelial cells (HSaVEC), human internal thoracic artery endothelial cells (HITAEC), and human adipose tissue-derived microvascular endothelial cells (HMVEC) after 24 h of culture in various endothelial cell media (EndoBoost, EndoBoost Plus, MesoEndo Growth, or EGM-2MV) and in a three-dimensional gel matrix recapitulating the composition of the endothelial basement membrane. Magnification: ×200. Scale bar = 100 µm.

**Figure 8 ijms-26-09217-f008:**
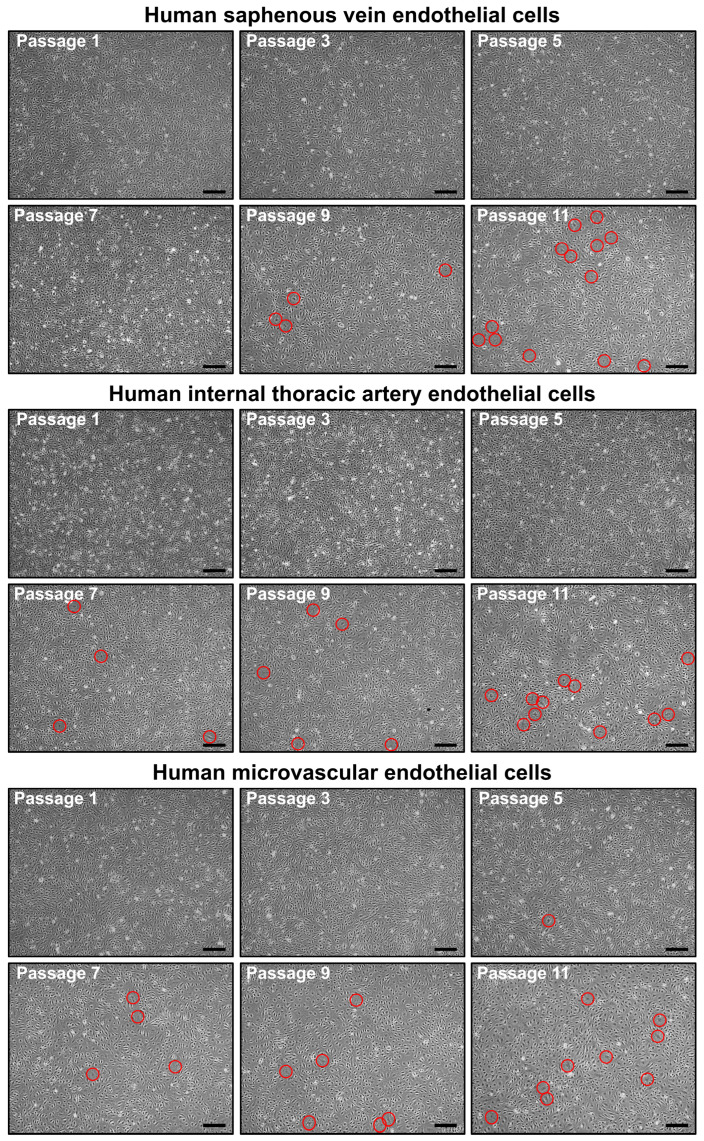
Phase contrast visualization of primary human saphenous vein endothelial cells (HSaVEC, top), human internal thoracic artery endothelial cells (HITAEC, middle), and human adipose tissue-derived microvascular endothelial cells (HMVEC, bottom) from passage 1 (starting from obtaining a pure culture) to passage 11. Note the increasing number of senescent cells (indicated by red circles) from passage 7 to passage 11. Magnification: ×200. Scale bar = 100 µm.

**Figure 9 ijms-26-09217-f009:**
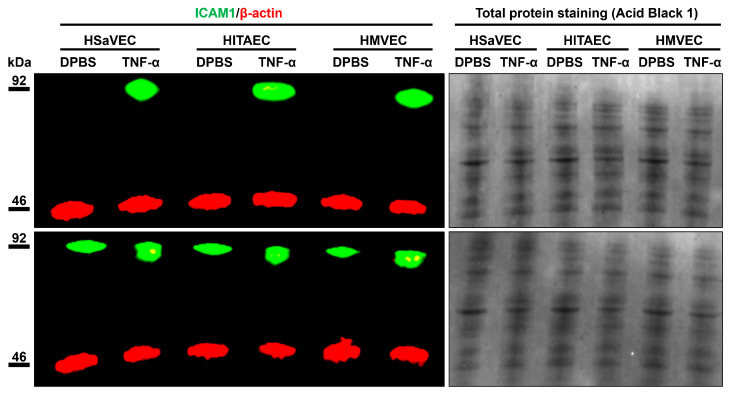
Measurement of intercellular cell adhesion molecule 1 (ICAM1) expression (green color) in mock (DPBS)- and TNF-α-treated primary human saphenous vein endothelial cells (HSaVEC), human internal thoracic artery endothelial cells (HITAEC), and human adipose tissue-derived microvascular endothelial cells (HMVEC) by fluorescent Western blotting. Red color indicates the expression of β-actin (a loading control). Total protein staining also shows an equal protein loading across the protein lysate samples. Molecular weight is indicated to the left of the images.

**Figure 10 ijms-26-09217-f010:**
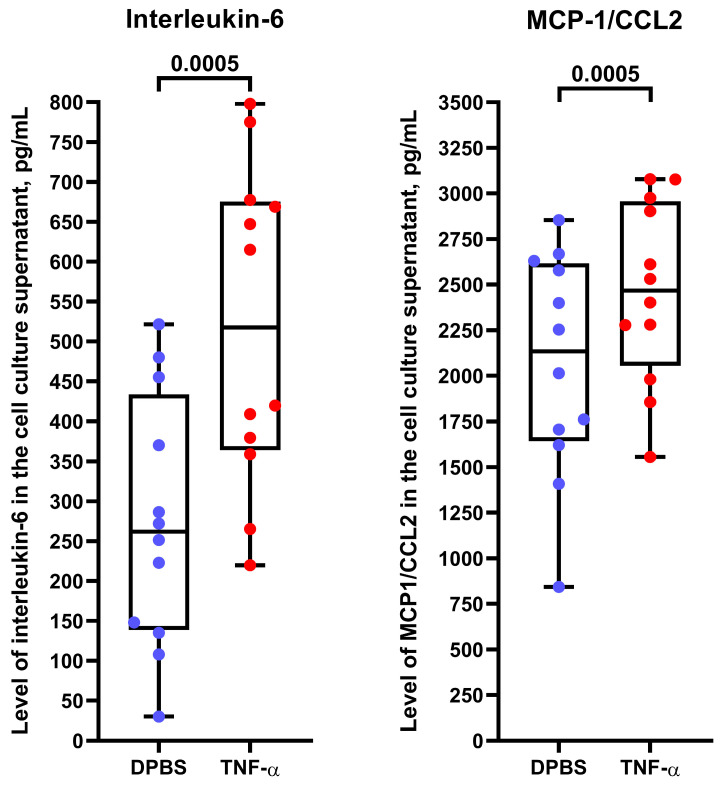
Enzyme-linked immunosorbent assay measurements of interleukin-6 and monocyte chemoattractant protein 1/C-C motif chemokine ligand 2 (MCP-1/CCL2) in the cell culture supernatant from TNF-α-treated primary human saphenous vein endothelial cells (HSaVEC), human internal thoracic artery endothelial cells (HITAEC), and human adipose tissue-derived microvascular endothelial cells (HMVEC). Blue and red dots indicate mock (DPBS)- and TNF-α-treated cells, respectively. Each dot on the plots represents a measurement from one patient (*n* = 4 patients per EC type, *n* = 12 patients per group). Whiskers indicate the range, box bounds indicate the 25th–75th percentiles, and center lines indicate the median. *p*-values are provided above the boxes, the Wilcoxon matched-pairs signed rank test.

**Figure 11 ijms-26-09217-f011:**
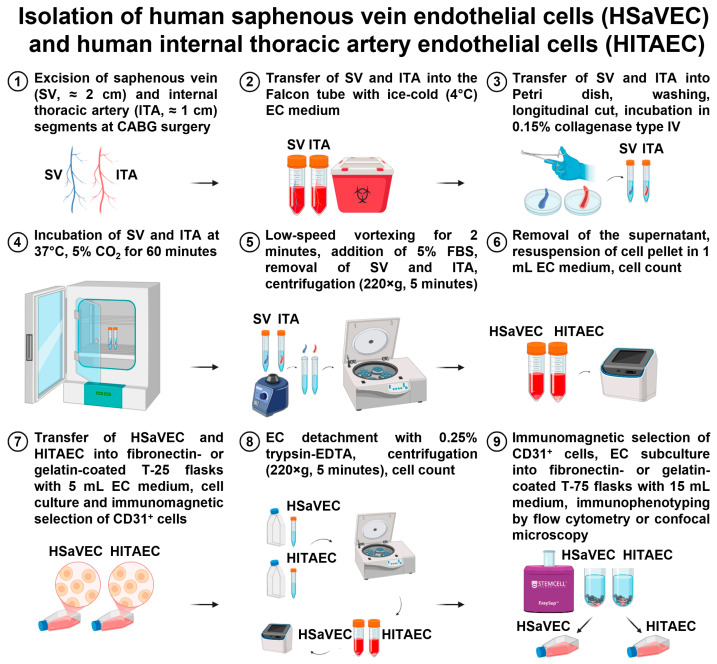
Isolation of primary human saphenous vein endothelial cells (HSaVEC) and human internal thoracic artery endothelial cells (HITAEC).

**Figure 12 ijms-26-09217-f012:**
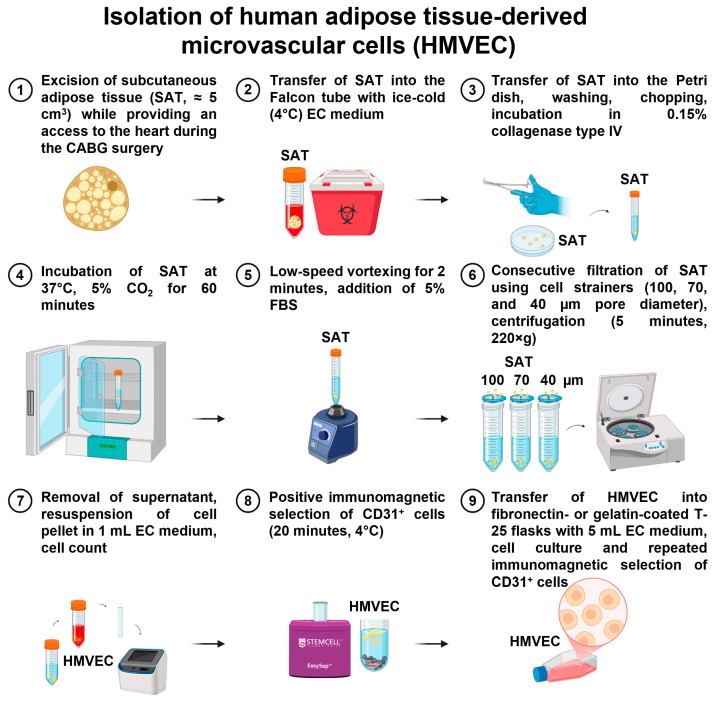
Isolation of primary human microvascular endothelial cells (HMVEC) from subcutaneous adipose tissue.

**Table 1 ijms-26-09217-t001:** Results of endothelial cell (EC) isolation and clinicopathological features of the patients who underwent coronary artery bypass graft (CABG) surgery. HSaVEC—human saphenous vein endothelial cells, HITAEC—human internal thoracic artery endothelial cells, HMVEC—human adipose tissue-derived microvascular endothelial cells, AH—arterial hypertension, DL—dyslipidemia, OB—obesity, DM—diabetes mellitus, CKD—chronic kidney disease, COPD—chronic obstructive pulmonary disease, CeVD—cerebrovascular disease, PAD—peripheral artery disease.

ID	Primary Endothelial Cell Lines			Comorbid Conditions									
	HSaVEC	HITAEC	HMVEC	Gender	Age	AH	DL	OB	DM	CKD	COPD	CeVD	PAD
1	+	−	−	male	69	+	+	+	+	+	−	−	−
2	+	+	−	female	75	+	+	+	+	+	−	−	−
3	−	+	−	male	58	+	−	−	+	+	−	+	−
4	+	+	+	male	55	+	−	+	−	+	−	−	−
5	−	−	−	male	52	+	−	+	−	−	+	−	−
6	+	+	+	female	72	+	−	−	−	+	−	+	−
7	−	−	−	female	74	+	+	−	−	−	−	−	−
8	−	−	−	male	68	+	−	+	−	+	−	−	−
9	−	−	−	female	78	+	−	−	−	−	−	−	−
10	+	+	+	female	63	−	+	+	+	+	+	+	−
11	+	+	−	male	72	+	−	−	−	+	−	−	−
12	−	+	−	male	59	+	−	−	−	+	−	+	−
13	+	−	−	male	71	+	−	+	+	+	−	−	−
14	+	+	−	male	62	+	+	+	+	−	−	+	−
15	+	+	−	male	67	+	−	−	−	−	+	−	−
16	+	−	−	male	59	+	−	−	−	−	−	−	+
17	−	−	−	male	64	+	−	−	−	+	−	−	−
18	+	+	+	female	61	+	+	−	+	−	−	−	−
19	+	−	−	male	64	+	+	+	−	−	−	−	−
20	+	+	−	male	60	+	−	−	−	+	−	+	+
21	+	−	−	male	74	+	+	−	−	−	−	−	−
22	+	+	+	male	59	+	−	−	−	−	−	−	−
23	−	−	−	male	67	+	−	−	−	−	−	−	+
24	+	−	−	male	70	+	+	−	−	+	−	−	−
25	+	+	−	male	69	+	−	−	+	+	−	−	−
26	−	−	−	male	73	+	−	−	−	−	−	−	−
27	−	+	−	female	73	+	−	+	+	−	−	−	−
28	+	+	−	male	67	+	−	−	−	+	−	+	+
29	+	−	−	female	76	+	+	−	+	+	−	+	+
30	−	−	−	male	70	+	−	−	+	−	−	−	+
31	+	−	−	male	63	+	−	−	−	+	−	−	−
32	+	+	−	male	72	+	+	+	−	+	−	+	−
33	+	+	+	female	73	+	−	−	+	−	−	−	−
34	−	−	−	male	75	+	−	−	−	−	−	−	−
35	−	+	−	male	74	+	−	−	−	−	−	−	−
36	−	−	−	male	73	+	−	−	−	−	+	−	+
37	−	−	−	male	57	+	−	+	−	+	−	−	−
38	+	+	+	male	64	+	+	−	+	+	−	+	−
39	+	−	−	male	73	+	+	−	−	−	−	−	−
40	−	−	−	male	67	+	−	−	+	+	−	−	+
*n*	24	19	7	31 ^1^	67.30 ^3^	39	13	12	14	21	4	10	8
%	60.00	47.50	17.50	77.50 ^2^	±6.51 ^4^	97.50	32.50	30.00	35.00	52.50	10.00	25.00	20.00

^1^ Number of males. ^2^ Proportion of male gender. ^3^ Arithmetic mean. ^4^ Standard deviation.

**Table 2 ijms-26-09217-t002:** Cultural features of primary human saphenous vein endothelial cells (HSaVEC), human internal thoracic artery endothelial cells (HITAEC), and human adipose tissue-derived microvascular endothelial cells (HMVEC) isolated from the patients who underwent coronary artery bypass graft (CABG) surgery.

Endothelial Cell Line/ Cultural Feature	Human Saphenous Vein Endothelial Cells (HSaVEC)	Human Internal Thoracic Artery Endothelial Cells (HITAEC)	Human Microvascular Endothelial Cells (HMVEC)
Approximate number of ECs in the colony	From 12 to 15	From 8 to 14	From 7 to 10
Approximate number of EC colonies per donor (in the T-25 flask)	From 15 to 17	From 8 to 10	From 3 to 5
Approximate number of positive immunomagnetic separations required to obtain an EC colony with ≥99% purity	From 1 to 2	From 3 to 4	From 5 to 6
Approximate number of days in culture (counted from the isolation) required for reaching confluence in T-25 flask	From 8 to 10	From 10 to 12	From 14 to 16
Approximate number of days in culture (counted from the isolation) required for reaching confluence in T-75 flask (i.e., at the second passage)	From 14 to 16	From 18 to 20	From 24 to 26

**Table 3 ijms-26-09217-t003:** Relative levels of gene expression (ΔCt, arithmetic mean and standard deviation, fold change, and *p* value) for the genes encoding pro-inflammatory cell adhesion molecules (*ICAM1*, *IL6*, and *CCL2*) in primary human saphenous vein endothelial cells (HSaVEC), human internal thoracic artery endothelial cells (HITAEC), and human microvascular cells (HMVEC) incubated with control Dulbecco’s phosphate-buffered saline (DPBS) or TNF-α for 24 h. Ratio paired t test (pairwise, patient-dependent comparison of ΔCt, *n* = 4 patients per EC type, *n* = 12 patients per group).

Gene	ΔCt, Arithmetic Mean	DPBS	TNF-α
	ΔCt, Standard Deviation		
	Fold Change		
	*p* Value		
*ICAM1*	ΔCt, arithmetic mean	0.1016	0.2756
	ΔCt, standard deviation	0.1095	0.3524
	Fold change	1.00	2.71
	*p* value	1.000	0.001
*IL6*	ΔCt, arithmetic mean	0.0655	0.2345
	ΔCt, standard deviation	0.1051	0.5619
	Fold change	1.00	3.58
	*p* value	1.000	0.002
*CCL2*	ΔCt, arithmetic mean	1.1670	2.6310
	ΔCt, standard deviation	0.7389	1.9120
	Fold change	1.00	2.25
	*p* value	1.000	0.002

**Table 4 ijms-26-09217-t004:** Primer sequences for reverse transcription-quantitative polymerase chain reaction.

Gene	Forward Primer	Reverse Primer
*ICAM1*	5′-TTGGGCATAGAGACCCCGTT-3′	5′-GCACATTGCTCAGTTCATACACC-3′
*IL6*	5′-GGCACTGGCAGAAAACAACC-3′	5′-GCAAGTCTCCTCATTGAATCC-3′
*CCL2*	5′-TTCTGTGCCTGCTGCTCATAG-3′	5′-AGGTGACTGGGGCATTGATTG-3′
*PECAM1*	5′-AAGGAACAGGAGGGAGAGTATTA-3′	5′-GTATTTTGCTTCTGGGGACACT-3′

## Data Availability

The original contributions presented in this study are included in the article. Further inquiries can be directed to the corresponding author.

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
