# Peer review of "Isolation of Primary Human Saphenous Vein Endothelial Cells, Human Internal Thoracic Artery Endothelial Cells, and Human Adipose Tissue-Derived Microvascular Endothelial Cells from Patients Undergoing Coronary Artery Bypass Graft Surgery"

_ijms, 2025, doi:10.3390/ijms26189217_

Round 1
Reviewer 1 Report
Comments and Suggestions for Authors
This manuscript presents data on human vascular endothelial cells of various origin simultaneously isolated from the vessels of 40 patients undergoing by-pass surgery. In multiple by-passes cases the use of both internal thoracic artery and autologous veins are employed and these vessels become the source for the primary EC isolation. Additionally, microvascular EC could be obtained using subcutaneous fat tissue or epicardial fat in small amounts. The authors describe the parallel isolation and propagation of all three types of EC from 6 of 40 patients undergoing CABG surgery. Methods of cell isolation, propagation and maintaining in culture are given in extensive detail which is the main value of the work. These methods are of great interest from the practical standpoint. The manuscript is supplemented with the broad literature outlook relevant to the potential applications of such EC cultures. Obviously, obtaining arterial, venous and microvascular EC from the same patient may be useful for further patient-oriented research and personalized clinical application.
Still, several deficiencies of the manuscript require improvement by the authors.
1. In Introduction the authors suggest that HSaVEC, HITAEC isolated by their approach can be further utilized in the studies exploring the pathophysiology of atherosclerosis and thrombosis (120-121). Whereas in the text above (96-100) they state that HITAEC and HSaVEC are derived from the vessels that are not subjected to atherosclerosis, stenosis and thrombosis. These two statements seem to contradict each other. How relevant are these cells then as an in vitro model system for atherothrombosis? Is it right that saphenous vein is resistant to thrombosis? Clinical evidence points to the contrary.
2. Abstract and Results. The metabolic (cell viability and proliferation) assay should be more correctly identified as cell viability and proliferation assay since no true metabolic parameters, for instance, respiration, lactate production or other metabolite conversion events were measured and presented in this work. It is obvious though that cell proliferation or any other manifestation of cell viability would require metabolic backup.
3. Fig. 1. EC were immunoidentified as CD31+CD146+CD90− to exclude CD90+ fibroblasts . However, CD90 (Thy-1) is present on activated EC. Were CD31+CD146+CD90+ cells present in EC primary isolate or at reselection stage?
4. Figure 4. Some HMVEC show only nuclear staining and no PECAM-1 and vWF signal (for instance, in the low left corner). How clean is this particular culture?
5. Figure 6. Measurements of bioimpedance indicating proliferation of primary human EC over 72 hours. Endothelial cells contribute to impedance by occupying the electrode surface as well as by establishing rather tight contacts with each other in contrast to many other cell types. Is it true proliferation or maturing contacts between EC, or both that are recorded in Fig.6 The state of EC confluency in the wells may assist with the answer.
6. Materials and Methods. Trypsin neutralizing solution (5% FBS) does not inhibit collagenase. The enzyme is rather stopped by dilution - centrifugation and cell resuspention.
7. Discussion is overstretched. Extended lists of basement membrane and matrix components (362-372) as well as of constitutive and inducible endothelial receptors (374-384) could be reduced or generalized. The issue of ECFC, although exciting, is peripheral to the main message of the manuscript and could be briefed / omitted.
Author Response
We sincerely thank the reviewer for the constructive criticism and valuable suggestions, which significantly helped us to improve the manuscript. Below we provide a point-by-point response to the reviewer’s suggestions. Please see the attachment.

Reviewer 2 Report
Comments and Suggestions for Authors
The authors investigated practical aspects of preparing primary cultures of three types of endothelial cells obtained from patients undergoing CABG, human saphenous vein endothelial cells (HSaVEC), human internal thoracic artery endothelial cells (HITAEC), and human microvascular endothelial cells (HMVEC). They assessed microscopic appearance, quantitative aspects of EC purification using immunomagnetic beads, and measures of cell proliferation and time to confluence. They also determined arteriogenic capacity of the cell lines. The idea is potentially valuable, since various commercially available endothelial cell lines differ in their ability to model aspects of atherosclerotic progression and vascular biology. However, the aspects of EC biology that the authors chose to study are puzzling. Why did they not assess relevant factors such as the ability of Yoda treatment to cause eNOS phosphorylation or TNF-alpha induction of ICAM-1? This idea was mentioned on page 13 and, in fact, an extensive list of markers and metabolites were listed, but none were assessed. Even more important, why did they not report the number of passages to senescence of these lines, especially since it would be extremely limited (possibly only 3 passages), considering the age of the donors, possibly nullifying the point of the exercise? One of the major reasons that HUVEC are so widely used for in vitro studies is that they provide about 9 useful passages, allowing expansion of early generations. In addition, why are these patients undergoing CABG when only 25% of them have CVD? Lines 39-47 are a run-on sentence.
Comments on the Quality of English LanguageSuboptimal English.
Author Response

(The authors gave the same response as above.)

Round 2
Reviewer 2 Report
Comments and Suggestions for Authors
I am impressed by how quickly the authors answered the comments with pertinent experimental information. My concerns have been satisfactorily addressed.